# Tribological Performance of Porous Ti–Nb–Ta–Fe–Mn Alloy in Dry Condition

**DOI:** 10.3390/ma13153284

**Published:** 2020-07-23

**Authors:** Carolina Guerra, Magdalena Walczak, Mamié Sancy, Carola Martínez, Claudio Aguilar, Marek Kalbarczyk

**Affiliations:** 1Departamento de Ingeniería Mecánica y Metalúrgica, Pontificia Universidad Católica de Chile (PUC), Santiago 7820436, Chile; caguerra2@uc.cl; 2Centro de Investigación en Nanotecnología y Materiales Avanzados, CIEN-UC, PUC, Santiago 7820436, Chile; mamiesancy@uc.cl; 3Escuela de Construcción Civil, Pontificia Universidad Católica de Chile, Santiago 7820436, Chile; carola.martinez@usm.cl; 4Departamento de Ingeniería Metalúrgica y de Materiales, Universidad Técnica Federico Santa María, Valparaíso 2390123, Chile; claudio.aguilar@usm.cl; 5Łukasiewicz Research Network, Institute for Sustainable Technologies, 26-600 Radom, Poland; marek.kalbarczyk@itee.lukasiewicz.gov.pl; 6Faculty of Mechanical Engineering, Kazimierz Pułaski University of Technology and Humanities, 26-600 Radom, Poland

**Keywords:** Ti-based alloy, porosity, X-ray diffraction (XRD), coefficient of friction, wear

## Abstract

The tribological properties of a novel porous Ti–Nb–Ta–Fe–Mn alloy with 0%, 30%, and 60% porosity were evaluated for biomedical applications. The tribotesting was performed using a ball-on-disc under dry conditions, using an alumina ball and 1 N of a load. The coefficient of friction at the early stage of the porous samples was lower than that of the bulk, 0.2 and 0.7, respectively, but the samples with 30% porosity shift toward the bulk value after a variable number of cycles, while the samples with 60% remained stable after 100,000 cycles. The wear rate of the specimen with 60% porosity was twice as low as that of the bulk. The results are explained by shift in wear mechanism associated with the modified bearing ratio of the porous surface and by the accumulation of wear debris inside the pores, which prevented the development of three-body abrasion.

## 1. Introduction

Ti-based alloys have been widely used in the orthopedic industry and today, as porous materials have gained particular interest due to the possibility of mimicking the elastic modulus (E) of natural bone and thus avoiding failures by stress shielding [1,2]. Nonetheless, the addition of porosity to the metal matrix may also modify other properties related to biocompatibility [3], corrosion [4], and wear resistance [5,6]. Some studies have shown that the corrosion resistance may decrease if porosity is added to Ti-based alloys [7]. However, in a previous study, the new Ti–Nb–Ta–Fe–Mn alloy was shown to remain passivated after 91 days of exposure to a simulated body fluid despite a high degree of porosity. In addition, no ion release was observed after high overpotential was applied, suggesting that a passive layer composed by Ti, Nb, and Fe protects the surface against corrosion [4]. On the other hand, regarding tribological properties, Ti-based alloys have been generally considered of poor resistance attributed to low capacity for work hardening [8], even when sliding against a softer material [1,9], requiring targeted alloying or coating. In this context, only a few studies have been focused on the tribological response of porous alloys. Liu et al. [10] reported the porosity-reduced resistance to wear of CP–Ti (commercially pure Ti), which was demonstrated by an increase in the coefficient of friction (COF) and wear volume. Whereas, Choi et al. [11] showed a considerable rise in the wear mass loss of Ti bulk over the sliding distance of 300–2000 m as compared with porous samples. Additionally, they studied the porous Ti-W alloy, reporting an improved wear resistance of the foam attributed to the incorporation W, which would increase the hardness of the alloy. Indeed, intermetallic compounds demonstrate generally superior mechanical properties as compared to solid solutions, increasing stiffness and thus improving wear resistance [12], although some specific types of intermetallic compounds might also be associated with lower machinability [13]. However, there is no agreement on the influence that porosity has on wear and friction performance, in particular the individual aspects of porosity percentage, pore shape, and size, for a given alloy composition [5,6].

Therefore, the present work aims at evaluating the tribological performance of a novel porous Ti–Nb–Ta–Fe–Mn alloy. Samples were prepared with a variation of porosity percentages; then, they were characterized by X-Ray diffraction, inspection of morphology as well as tribological testing under dry conditions. Clarifying the effect of porosity on tribological performance offers an opportunity for biomedical applications for which the alloys has already been shown of interest for its mechanical properties and corrosion resistance.

## 2. Experimental 

### 2.1. Preparation of Samples 

The Ti–20Nb–11Ta–16Fe–1Mn (at.%) composition was obtained by mechanical alloying in a planetary mill. The bulk sample (considered as 0% porosity) was obtained by Spark Plasma Sintering (SPS) at 1000 °C and 60 MPa for 7 min holding time. The 30% and 60% porosity samples were obtained by the powder metallurgy (PM) technique using ammonium bicarbonate (NH_4_HCO_3_) as a space holder. The materials were compacted at 430 MPa and then sintered in two steps (1.5 h at 180 °C followed by 3 h at 1300 °C) and cooled at the rate of 5 °C‧min^−1^ as described in Ref. [4]. 

### 2.2. Material Characterization

Chemical composition of the sintered alloys was conducted using elemental mapping (EM) and dispersive X-Ray spectroscopy (EDX) coupled to an Field Emission Scanning Electron Microscope (FE-SEM, QUANTA FEG, Santiago, Chile) 250 in high vacuum (3.7·10^−4^ Pa). Phase composition was characterized by X-ray diffraction (XRD), using a multipurpose powder diffractometer (STOE STADI MP, Santiago, Chile), equipped with a Cu–K*α*_1_ radiation source (*λ* =1.54056 Å, curved Germanium (111) monochromator of the Johann-type) and a microstrip detector DECTRIS MYTHEN 1K. The phases were identified using MATCH! software (Crystal Impact, Bonn, Germany) [10] version 3.9.0.158 equipped with the Crystallography Open Database (COD). Electrochemical characterization and data on corrosion products can be found in Ref. [4]. 

### 2.3. Tribological Characterization 

For the tribological testing, the samples were mounted in a thermosetting resin, cut, and polished to obtain a suitable surface of low roughness; then, they were cleaned using isopropyl alcohol. A ball-on-disc tribotester (NTR2 nano tribometer, CSM Instruments) was used, set to oscillating mode, and operated at 20 ± 1 °C under technically dry conditions. As counter specimen a Al_2_O_3_ balls of 2 mm of diameter were used, applying 1000 mN of load with stroke of 0.6 mm (0.3 mm amplitude). The maximum linear speed was 1 cm‧s^−1^, and the test was run for 100,000 cycles. Ball material and test parameters were selected on the basis of preliminary tests. The wear tracks of tested specimens (three per sample) were analyzed using an optical profilometer (Talysurf CCI-Lite Non-contact 3D profiler), extracting three profiles from each sliding path: one central and two adjacent (100 µm from the central to each side). 

## 3. Results and Discussion

### 3.1. Morphological and Microstructural Characterization

Figure 1a,b exemplify the microstructure and phase composition of the 30% porosity sample. The top view FE-SEM images reveal the interconnected porous structure produced during sintering, where two types of pores are apparent: the fine pores (approximately 5 µm) inherent in the manufacturing method, and larger pores (approximately 450 µm) produced by removal of the space holder, as indicated in the figure. Figure 1c shows the elemental contrast by a high magnification cross-section FE-SEM image, revealing a bi-modal composition of darker and lighter regions corresponding to Ti-rich and Nb–Ta-rich phases. The overall distribution of the alloying elements in Figure 1b showed a homogenous distribution of the alloying elements with a relatively low content of Mn, which is indicative of the solid-state reaction was achieved. On the other hand, analysis of the X-ray diffraction pattern in Figure 1d reveals the presence of Ti compounds. In the peak around 38.5°, the contribution of two *β*-type phases (cubic, Im3m) are found, which are identified as TiNb and TiTa. In addition, four compounds are associated with the remaining peaks, among them TiFe (cubic, Fd3m), TiO_2_ (cubic, Fm3m), Nb_0.3_Ti_0.7_ (orthorhombic, Cmcm), and FeNb (rhombohedral, R3m). Thus, the alloy consists mainly of the *β*-phase, which is known to provide stiffness and ductility without compromising the mechanical strength of a Ti alloy [1], whereas intermetallic compounds, such as the Nb_0.3_Ti_0.7_, would provide hardness to the softer matrix, reinforcing the alloy [12,14,15].

### 3.2. Friction Performance

Figure 2 summarizes the evolution of COF during the tests, revealing that the friction response differs for the bulk and porous materials. For the bulk and 60% porosity samples, a stable COF is observed, with the average values of 0.78 and 0.22, respectively. The 30% porosity sample experienced a shift of COF from 0.2 to 0.75 after a variable duration of the early stage, indicated by an arrow in Figure 2b.

The COF at steady state registered for bulk and 30% porosity samples is similar to that of bulk Ti-6Al-4V, which is typically reported around 0.7 for comparable test conditions [16,17]. This observation indicates that the shift in the COF of the 30% porosity sample is associated with the change of friction mechanism, which is possibly associated with the accumulation of wear debris inside the pores. Such an effect was shown by other authors for a Ti-6Al-4V coating, in which the porosities percentage and pore diameters are approximately 13% and 25 μm, respectively, contributed to the accumulation of debris by the pore-filling mechanism [5]. The randomness in response of the sample with 30% porosity is attributed to variability of number of pores that are included in the initial contact area. This interpretation would also explain the prolonged steady state friction of the sample with 60% porosity that can absorb more debris. A similar result has been observed in TiNbC coatings, where debris was stacked at the bottom of the wear track, not participating in the process [18].

### 3.3. Wear Performance

Figure 3 shows representative 3D images and center cross-sections of the sliding track. The track on the 60% sample is the narrowest and most shallow (note the scale of the depth axis). The average depth of the wear track on the bulk and 30% samples is similar, but the cross-section area for the 30% sample is higher, also revealing large pores. Since the surface selected for the test was free of large pores before testing, these pores must have emerged during the test. 

The wear tracks for the bulk and 30% porosity samples reveal that the alumina ball penetrated deeper than in the 60% porosity sample, in case of which, after 100,000 cycles, the ball hardly penetrated the most porous surface. This observation is indicative of a distinctive wear system that developed between the two bodies, in which both the effective surface of contact and the debris generated in the process would play a different role. In addition, the amount of surface oxides present in the interior of the sintered structure should be taken into account, as it is known to be potentially detrimental when released as debris [5,6]. Then, the lesser penetration in the more porous sample is indicative of debris removal. Such an effect was observed by Dubrujeaud et al. [6] in Distaloy AE of various porosity tested in dry conditions. Their explanation was that the debris, mainly iron oxide particles in this case, was trapped inside the pores. Moreover, the effect was most notorious for pores larger than 12 µm. For smaller pores, the debris would be participating in the wear process as a third body, leaving wear tracks similar to those shown in Figure 3a for the 30% porosity.

Table 1 summarizes the numerical values of the wear results and selected functional parameters determined from Abbott–Firestone curves [19,20] measured prior tribotesting: core roughness depth (S_k_), which represents the efficient roughness, and the areal height difference (S_dc_) calculated for the upper 50% of surface points. The high value of S_k_ is associated with irregularities susceptible to shear stress, and the generation of hard debris enhancing the “third body” abrasion mechanism, i.e., a system composed of the ball, the sample, and the debris. The S_k_ value for the 30% porosity sample is higher than the other, demonstrating that it was non-uniform and rougher, compared with the others, and therefore, it would be more susceptible to wear as discussed by Czyrska-Filemonowicz et al. [21]. In general, the higher bearing ratio is related to the larger contact area [22]. Although the test load was the same in each test, it resulted in higher contact pressures for samples with lower bearing ratio, thus raising the wear intensity. The 60% porosity sample provides a higher bearing ratio, and because the pores are small and shallow, the S_dc_ value is lower as compared with the 30% porous sample. In the case of 30% porosity, new pores emerge during the wear progress, limiting the bearing ratio and reducing the contact area, leading to higher contact stresses and enhanced wear intensity, as is observed also through S_dc_ and S_k_ [20]. 

The elastic modulus decreases as a function of increasing porosity percentage, achieving values around 6 GPa for the higher porosity sample [2]. The wear rate of the sample with the highest porosity was twice less than that of the bulk, which is related to the type of abrasion, where in the first one, a two-body abrasion occurred, while in the bulk, there was three-body abrasion [23]. However, it is expected that when the sample with 60% had its pores filled with debris, the wear rate and COF increase.

On the other hand, considering the phase composition of the alloy, the good resistance of the bulk metal to wear can be expected, owing to a combination of low elastic modulus and effective hardness provided by the presence of intermetallic compounds, as explained by other authors [12,24]. In addition, the presence of niobium oxide (Nb_2_O_5_) formed inside the pores during sintering [4] can be expected to provide lubrication, which is beneficial for lowering the wear of implants, as discussed by Dinu et al. [25]. 

Figure 4 shows the appearance of the Al_2_O_3_ balls, which reveals the effective contact area after 100,000 cycles. This contact area is significantly lower for the 60% porosity sample, which is in agreement with the observed above by COF and wear track. Although the contact area is similar for bulk and 30% porosity sample, the mechanism of wear is different. Whereas scars and grooves are observed for the 30%-Al_2_O_3_ pair, layers of material are visibly transferred from the sample to the ball [6]. These observations are indicative of abrasive and adhesive wear mechanisms [9,16], respectively. 

Such a difference in wear mechanism may be caused by a dissimilarity in the relative hardness or by the appearance of the wear products at the interface between the target alloy and the counter surface, i.e., the test ball. Since the hardness of the sintered matrix can be expected to be similar, the difference should be attributed to the debris, as observed by Dubrujeaud et al. [6] and Munagala et al. [5]. For the bulk sample, at the early stage, abrasive wear is likely to set in, but after accumulating plastic deformation, a debris was evidently generated and transferred to the surface of the ball, which is consistent with the observation of Haftlang et al. [26]. However, the wear mode of the 30% porosity sample is dominated by the three-body mechanism, consisting of the target surface, the ball, and the unrestricted debris, which can be deduced from the deep and marked wear tracks at the ball surface. On the other hand, the 60% porosity surface paired with the Al_2_O_3_ ball has shown the least damage of a type characteristic to the two-body, which corroborates with the conclusion that the debris must have been absorbed by the pores in this case. 

## 4. Conclusions

The highly porous Ti–Nb–Ta–Fe–Mn alloy was shown to have a superior tribological response as compared to a bulk sample, exhibiting a consistently lower COF. The wear rate was twice as low as that of the bulk sample and stable in time. In the case of intermediate porosity, the COF was significantly lower, but after an initial period of variable duration, it increased to the value of bulk coefficient. Both friction and wear behavior are mainly explained by the bearing ratio of the porous materials and the debris generated during testing. The debris can be lodged in the pores, modulating the friction and wear responses. On the other hand, the combination of β-titanium and the presence of intermetallic phases provided mechanical properties that favor resistance to wear. However, once the pores are filled with the wear debris, the COF increases, and large pores are generated in association with increased contact pressures. Since the pores may collaborate in delaying the wear of the porous Ti-based alloy, a beneficial tribological performance can be expected in application for joint replacement. 

## Figures and Tables

**Figure 1 materials-13-03284-f001:**
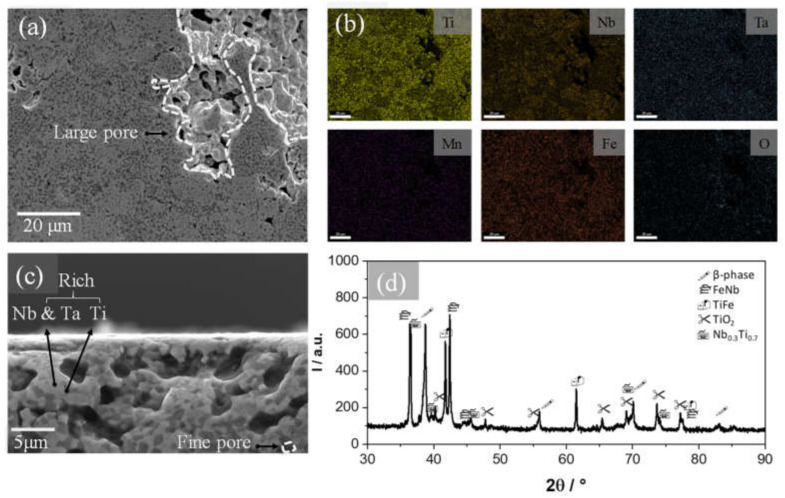
Microstructure and composition of as-sintered 30% porosity sample: (**a**) Top view FE-SEM micrograph, (**b**) Elemental mapping of the alloying elements in image (**a**), (**c**) Cross-section FE-SEM micrograph, and (**d**) XRD pattern with peaks marked with the corresponding phases.

**Figure 2 materials-13-03284-f002:**
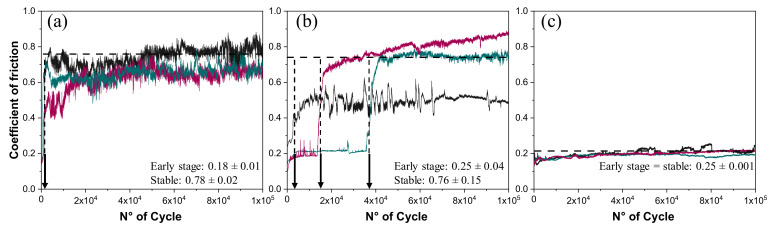
Evolution of coefficient of friction (COF) during test: (**a**) bulk sample, (**b**) 30% porosity, and (**c**) 60% porosity samples. Each curve is an independent measurement. The arrows indicate the duration of the early stage.

**Figure 3 materials-13-03284-f003:**
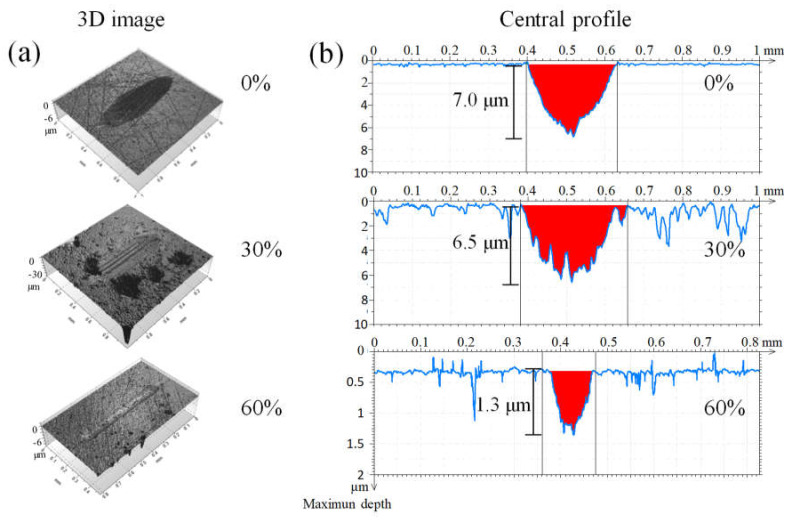
Images of wear tracks on the surface obtained by (**a**) optical profilometer for samples with different porosity and (**b**) the scratch profile after being tested.

**Figure 4 materials-13-03284-f004:**
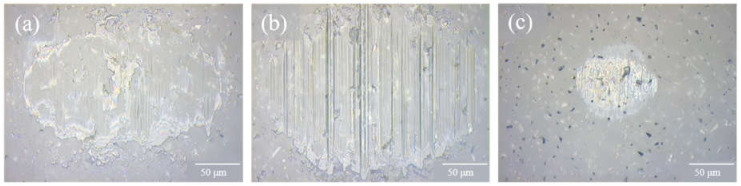
Optical images of the contact areas observed on the alumina balls after completing 100,000 test cycles on (**a**) bulk, (**b**) 30%, and (**c**) 60% porosity samples.

**Table 1 materials-13-03284-t001:** Summary of wear data and functional parameters.

ID/%	E/GPa	Max. Depth/µm	Cross-Section Areas/µm^2^	S_k_/µm	S_dc_/µm	Wear Rate/m^3^·N^−1^·m^−1^
bulk	48.8 ± 19.2	6.3 ± 1.2	856.6 ± 204.8	0.07	0.08	6.0 ± 2.0·10^−7^
30	8.8 ± 2.1	5.4 ± 2.6	1111.0 ± 130.1	0.50	0.81	5.5 ± 4.2·10^−7^
60	5.7*	0.6 ± 0.3	28.8 ± 0.2	0.07	0.16	2.7 ± 1.2·10^−7^

* Estimated by Gibson–Ashby relation [16].

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
