# Peer review of "Tribological Performance of Porous Ti–Nb–Ta–Fe–Mn Alloy in Dry Condition"

_materials, 2020, doi:10.3390/ma13153284_

Round 1

Reviewer 1 Report

Dear editors and authors,

In general the idea and paper is quite okay however, there are things still needs to be improved, which is as follows:
1. In the Introduction section, please add reference in "modify other properties related to biocompatibility, corrosion, and wear resistance".
2. ALso in the Introduction section, to help reader, please give a brief explanation, why does "Ti-based alloys have been generally considered of poor resistance [5]"
3. In the Introduction section (page 2 before the last paragraph of this section).
The authors mentioned "Indeed, intermetallic compounds have demonstrated superior mechanical properties than ordinary metals, which..."
I understand in this context (or specific applications) it may be true, but giving a general statement like this is a bit misleading because intermetallic particles can also be harmful to mechanical properties (see DOI: 10.5772/intechopen.73188 for example). Please rephrase this statement.
4. In the last part paragraph of Introduction, the authors should add briefly, what would be the benefits of this work?
5. In Section 2.2, line 3, there are typos, for instance High Vacuum. If this is not a brand it should not be capitalized (please double check). Please also double check other things related to typos.
6. In Section 3.1, last line in page 2, the symbol after the fine pores (~5... shows an error in the pdf. In page 3, still in the same section, also after "larger pores (~450.... also after ""the contribution of two phases ..-type (cubic, Im3m)...please double check again for the entire manuscript.
7. In Section 3.2, in the paragraph below Figure 2, authors mentioned "which is typically reported around 0.7 for comparable test conditions..". In Fig 2b, there is one sample (black line) which has COF value increase the earliest and then stays at around 0.5...can authors comment on this result? Any references?
8. Also in the same paragraph as #7, authors mentioned "change of friction mechanism, possibly associated with the accumulation of debris in the pores." Could authors provide any references for low porosity behavior on debris?
9. Also in the same paragraph as #7, why the authors did not provide any remarks, explanation and discuss about the cycle number where COF of the samples start to increase (black arrows in Fig. 2b) ?
10. In the first paragraph of Section 3.3 of page 4, why the authors only described results. There are no explanation and discussion in for this result. Please add some explanation and discussion along with references.
11. In page 5 above Table 1, authors mentioned "bearing ratio, thus rising wear intensity. The 60% porosity sample provides a higher bearing ratio, and because the pores are small and shallow, the Sdc value is lower compared with the bulk. Thus, the pace of bearing ratio rises. In the case of 30% porosity, new pores emerge during the wear progress, limiting the bearing ratio and reducing the contact area, leading to higher contact stresses and enhanced wear intensity."
But there's no comparison with other work and discussion about this. Can authors please add these ?
12. In page 6, on the paragrph above Figure 4, authors mentioned "...the mechanism of wear is different." but there are no explanation how different is the mechanisms. Please add here.
13. In the Conclusion section, I believe this section is too long and provide too much discussion (e.g. some important discussion are written here e.g. ...combination of beta-titanium and presence of intermetallic phases..., which I believe important and not really described properly in the Result and Discussion section). Please move accordingly.

I hope the review is useful. Thank you for your attention.

Reviewer 2 Report

The paper represents a significant study of the tribological response in the porous alloy. It is a significant topic since its important application in the orthopedic industry. Considering the format of this paper is a short communication, it is up to standard for publication. I think, nonetheless, that the manuscript could be improved if the authors could address the comments and recommendations I listed below.

  1. The novelty of this research should be highlighted in the Abstract.
  2. At the end of page 2, the unit of your fine pores is wrong(or your typo). 
  3. In figure 1a, Could you mark a representative fine pore and larger pore? The upper right region is the large pores or sample defect?
  4. I highly recommend you substitute the markers in Figure 1c to more formal markers like a triangle, square...
  5. Figure 2 is hard to read. Could you change the curves to different colors?  
  6. In figure 2c, if I understand correctly, the early stage of 60% porosity sample is instantaneous. How you got the COF value of the early stage? Based on your figure, the curve never goes above 0.25.
  7. In the content below figure 4. What is the three-body, please explain or give a reference. 
  8.  Your conclusion section should be concise and to the point. The second paragraph looks like a discussion of mechanism. 
  9. In the introduction part, you have some discussion about corrosion and passivation. I'm interested to see the electrochemical polarization curve of your samples. Give some study of the passivation or oxidation of your samples. It could be helpful to have a better understanding of the mechanism.

Round 2

Reviewer 1 Report

Dear authors, thank you very much for responding to my review properly. It is a good journal. I wish you all the best.